# The Genetics of Fitness Reorganization during the Transition to Multicellularity: The Volvocine *regA*-like Family as a Model

**DOI:** 10.3390/genes14040941

**Published:** 2023-04-19

**Authors:** Zachariah I. Grochau-Wright, Aurora M. Nedelcu, Richard E. Michod

**Affiliations:** 1Department of Biology, The College of New Jersey, Ewing, NJ 08628, USA; 2Biology Department, University of New Brunswick, Fredericton, NB E3B 5A3, Canada; 3Department of Ecology and Evolutionary Biology, University of Arizona, Tucson, AZ 85721, USA

**Keywords:** multicellularity, cellular differentiation, life history, individuality, gene co-option, *Volvox*, *Chlamydomonas*, volvocine algae, *regA*, SAND domain

## Abstract

The evolutionary transition from single-celled to multicellular individuality requires organismal fitness to shift from the cell level to a cell group. This reorganization of fitness occurs by re-allocating the two components of fitness, survival and reproduction, between two specialized cell types in the multicellular group: soma and germ, respectively. How does the genetic basis for such fitness reorganization evolve? One possible mechanism is the co-option of life history genes present in the unicellular ancestors of a multicellular lineage. For instance, single-celled organisms must regulate their investment in survival and reproduction in response to environmental changes, particularly decreasing reproduction to ensure survival under stress. Such stress response life history genes can provide the genetic basis for the evolution of cellular differentiation in multicellular lineages. The *regA*-like gene family in the volvocine green algal lineage provides an excellent model system to study how this co-option can occur. We discuss the origin and evolution of the volvocine *regA*-like gene family, including *regA*—the gene that controls somatic cell development in the model organism *Volvox carteri*. We hypothesize that the co-option of life history trade-off genes is a general mechanism involved in the transition to multicellular individuality, making volvocine algae and the *regA*-like family a useful template for similar investigations in other lineages.

## 1. Introduction

The evolution of multicellularity is the premier example of an evolutionary transition in individuality, which involves the integration of lower-level individuals (cells, in this case) into a new kind of higher-level individual (a multicellular organism). Other examples of evolutionary transitions in individuality are the integration of networks of cooperating genes into the genome of the first cell, the origin of eukaryotic cells, and the evolution of eusocial insect societies [1,2,3]. As an evolutionary transition in individuality, the evolution of multicellularity is especially interesting because it occurred repeatedly throughout the tree of life (in both prokaryotes and eukaryotes) and at vastly different points in the history of life [4,5]. Multicellular phenotypes can be either stable/obligate or facultative (i.e., in response to environmental cues, as a part of the life cycle) and involve clonal (e.g., in animals, plants, algae, or fungi) or non-clonal (e.g., in social amoebae or myxobacteria) groups, with or without cell specialization [5,6]. Multicellularity can also evolve in the lab and can be studied using experimental evolution [7,8,9,10,11].

For billions of years, life on Earth was composed of solely single-celled individuals. At different points in the history of life, cells formed groups, and these cell-groups evolved into stable, integrated multicellular individuals with heritable variation in fitness at the level of cell-groups. However, how groups of cells evolve into a new kind of individual is not well understood. We have previously proposed that the transition of a group of cells into a multicellular individual requires the reorganization of fitness from being a property of single cells to being a property of the multicellular group [12,13].

To envision the reorganization of fitness during the transition to multicellular individuality we need to consider the general properties of fitness, its components, and how these fitness components are trade-offs for one another. The fitness of any evolutionary individual involves two basic components: reproduction and survival. High overall fitness requires a balance between these components; in the simplest mathematical model, fitness is taken as the product of survival and reproduction (for example, [14]). Nevertheless, survival and reproduction are often traded off for one another, especially in limiting conditions, such that investment in one fitness component tends to detract from the other.

During the transition from unicellular to multicellular individuals, the activities associated with reproduction and survival, initially expressed in a way that enhances the fitness of individual cells, become reorganized among the cells of the multicellular group. Ultimately, cells differentially express one or the other component (i.e., cells become specialized) to increase the fitness of the new multicellular individual. This division of labor between cells in the multicellular group reorganizes the two fitness components, reproduction and survival, from the cell level to the group level. Once the fitness components are compartmentalized between cells specialized in reproduction (germ) and survival (soma), the multicellular group becomes a multicellular individual; that is, it is indivisible in the sense that the specialized cells lose their own individuality and can no longer both survive and reproduce outside of the context of the group [12,14,15].

How can fitness become reorganized during an evolutionary transition in individuality? In particular, what is the genetic basis underlying the reorganization of fitness resulting in cell specialization in fitness components during the evolution of multicellularity? Understanding the relative contribution of the two fitness components throughout the life of all individuals can provide a mechanistic framework. Many life history traits control resource contributions to survival and reproduction (or to distinct aspects of survival and reproduction, e.g., growth vs. maintenance or number vs. quality of offspring) as the individual proceeds through its life cycle. Furthermore, in times of stress, or when resources are limited, overall fitness is compromised, and individuals must commit to survival and delay reproduction until conditions improve. All organisms must possess genes that decrease reproduction in times of stress to ensure survival. Genes that control investment in fitness components in times of stress are examples of life history genes. Our hypothesis is that genes that turn off reproduction in times of stress in unicellular organisms were co-opted during the evolution of multicellularity to produce non-reproductive somatic cells in cell groups [16,17].

Consistent with our hypothesis, current evidence indicates that de novo gene evolution and gene family expansion played a minor role in the evolution of multicellularity and cellular differentiation in both the animal and green algal/plant lineages. Specifically, many genes that are involved in traits associated with multicellularity (and that were initially thought to have evolved during the evolution of multicellularity) did in fact evolve from genes present in their unicellular ancestors, wherein they played roles associated with the unicellular lifestyle. Thus, gene co-option (often involving gene duplication followed by diversification) appears to have been a major contributor to the evolution of cellular differentiation [18,19]. Such a scenario can be especially envisioned for the evolution of soma and germ, the two components that embody the full transition to multicellular individuality.

Studying such gene co-option events are challenging in many multicellular lineages due to the ancient origin of multicellularity and lack of closely related unicellular and simple multicellular relatives. For example, metazoans evolved multicellularity 574–852 million years ago [20], and lineages resembling transitionary ancestors are not known. However, a group of green algae in the order of Volvocales provides an ideal model system for studying the transition from unicellular to differentiated multicellularity. This group—known as the volvocine algae—evolved multicellularity relatively recently (~240 million years ago) and contains extant relatives that span a range of complexities from unicellularity, to undifferentiated multicellularity, to differentiated multicellularity [21,22,23] (Figure 1). Furthermore, consistent with our hypothesis, the evolution of somatic cell differentiation in this lineage involved the co-option of a stress-induced life history gene that belongs to the *regA*-like gene family [12,17,24], which is the subject of this review. As this gene family is specific to volvocine green algae and not found in other lineages, our hope is that the research reviewed here may stimulate efforts to identify similar genes in other lineages. We predict that such work would reveal additional examples of life history genes that have been co-opted for fitness reorganization and the evolution of soma during the transition to multicellular individuality in other groups.

## 2. The Volvocine Model System

The volvocine lineage is a group of freshwater haploid bi-flagellated chlorophyte green algae that reproduce asexually in optimal environments but can undergo rounds of sexual reproduction under stressful conditions [26]. This group has been developed as a model system for the evolution of multicellularity and cellular differentiation because its species span a range of morphological and developmental traits from single-celled organisms (e.g., *Chlamydomonas*), to multicellular forms without cell specialization (e.g., *Gonium* and *Eudorina*), to multicellular organisms with complex embryonic development and germ–soma differentiation (i.e., *Volvox*) [23,27,28] (Figure 1).

The multicellular volvocine species are included in three families: Tetrabaenaceae, Goniaceae, and Volvocaceae. In addition, within the Volvocaceae family, two sub-clades have been defined: the “Eudorina Group” and the Euvolvox (or section Volvox) [29] (Figure 2). The Tetrabaenaceae family contains two species, *Tetrabaena socialis* and *Basichlamys sacculifera*. These are the simplest multicellular volvocine algae with four *Chlamydomonas*-like cells arranged like a four-leaf clover [30,31]. The polyphyletic Goniaceae family includes several species in two genera, *Gonium* and *Astrephomene*. The *Gonium* species have 8–16 *Chlamydomonas*-like cells arranged as flat plates, whereas the *Astrephomene* species are 32- or 64-celled spheroidal colonies with 2 to 4 sterile somatic cells in the posterior of the colony [26,32,33]. The Volvocaceae is the largest and most diverse family of volvocine green algae with many polyphyletic genera. Algae in the genera *Eudorina, Pandorina, Volvulina, Yamagishiella,* and *Colemanosphaera* all have spheroidal body plans with between 16 and 64 cells (cell numbers vary between genera) with no germ–soma cellular differentiation under standard growth conditions. Species in the *Pleodorina* genus have 32 to 128 cells with specialized somatic cells in the anterior portion of the colony, except for one species, *Pleodorina sphaerica*, that has somatic cells distributed in both the anterior and posterior of the colony [26,34]. Finally, species in the genus *Volvox* are the largest and most complex members of the Volvocaceae, with several hundred to several thousand cells and two distinct cell types, specialized germ and specialized soma [26].

Multicellular volvocine algae evolved from unicellular ancestors related to species in the *Chlamydomonas* and *Vitreochlamys* genera. Historically, analyses based on single or small sets of genes have indicated that multicellularity has arisen only once in the volvocine green algae clade [22]. However, a recent phylotranscriptomic analysis that more than quadrupled the number of single-copy nuclear genes used for phylogenetic reconstruction suggests that (i) multicellularity possibly evolved twice in this group, and (ii) the Goniaceae family is not monophyletic [35]. While this conclusion requires additional confirmation, we adopt this new phylogenetic hypothesis as the evolutionary framework for this review. The current tree topology (Figure 2) also implies that cellular differentiation independently evolved four to six times in volvocine green algae, which is consistent with past analyses [25,27].

The species *V. carteri* forma *nagariensis* has served as the primary model organism for studying the developmental and genetic mechanisms that underlie cellular differentiation [23,36,37]. An asexual *V. carteri* individual contains 1000–2000 small *Chlamydomonas*-like flagellated somatic cells and up to 16 large unflagellated germ cells known as gonidia (Figure 1). The somatic cells are terminally differentiated and have no cell division potential. The lack of cell division ensures that the motility of the individual is maintained because flagellar activity is compromised during cell division in volvocine algae due to the so-called “flagellation constraint” and the presence of a rigid cell wall [38]. Juvenile *V. carteri* develop from gonidia, which grow to up to ~1000 times the volume of somatic cells before they start dividing. The development of a juvenile *V. carteri* begins with a series of 5 symmetric divisions resulting in a 32-celled embryo. Then, during the sixth division cycle, the sixteen cells in the anterior of the embryo divide asymmetrically with one daughter cell inheriting a larger volume of cytoplasm than the other. These large cells go through 2 additional asymmetric divisions and then cease dividing, while all other cells go through a series of 11 to 12 symmetric divisions. At the end of cleavage, the embryo contains ~2000 cells, most of which are small soma initial cells, except the 16 large germ cell initials generated through asymmetric division. At this stage, the embryo is effectively inside-out relative to the adult organization, with the flagella of the somatic cells pointing inward. To gain the adult configuration, the embryo goes through an inversion process.

Cytodifferentiation occurs just after inversion; all cells of <8 µm terminally differentiate into somatic cells, while larger cells become germ cells [37]. Cell size has been shown to be sufficient for determining cell fate in *V. carteri* [39], though this is not the case in all *Volvox* species [40,41,42,43]. In *V. carteri*, a gene known as *regA* is turned on in small cells, which results in the suppression of germ cell development and the differentiation of somatic cells. On the other hand, a set of *lag* genes are thought to be specifically induced in large cells, which suppresses somatic cell development and initiates the differentiation of germ cells [37]. Currently, the presence of *lag* genes has only been established through mutant phenotypes and linkage mapping, while the actual genes themselves are unknown. Several other developmental genes (involved in asymmetric cell divisions and inversion) have been identified in *V. carteri* [37] but are beyond the scope of this review. Herein, we focus on *regA* and its family as major players in fitness reorganization between soma (survival) and germ (reproduction). Below, we first describe the main characteristics of the *regA* gene in *V. carteri* and then expand our discussion of the broader VARL gene family.

## 3. *regA* Gene Structure and Function

Early investigations into cellular differentiation in *V. carteri* identified a class of mutants called “somatic regenerators” in which somatic cells appear to first develop normally but then dedifferentiate and become reproductive [44,45] (Figure 1f). Linkage analysis found that all such regenerator mutants map onto a single locus which was named *regA* (from “regenerator”) [36,46,47]. Huskey & Griffin [47] originally described a second *regB* locus based on linkage group analysis of regenerator mutants, but reexamination of *regB* mutants by members of the same research lab determined that they are not regenerator mutants and have a different mutant phenotype [48]. Thus, in retrospect, all regenerator mutants can be mapped onto the *regA* locus [36]. However, it is worth noting that the annotation “RegA” or “Reg genes” has been used multiple times independently in species from other groups (e.g., bacteria and animals) to refer to different genes coding for distinct unrelated proteins. Such similarities in name are due to historical and linguistic coincidence rather than any shared function or homology. In this review, we are strictly discussing the *regA* gene and its gene family that is restricted to volvocine algae.

Based on the link between somatic regeneration and the *regA* locus, the *regA* gene was deemed the master regulatory gene that controls somatic cell development in *V. carteri* [36,37,49]. Kirk et al. [49] used transposon tagging to identify the *regA* gene and went on to determine that the RegA protein is localized in the nuclei of somatic cells. In *V. carteri* f. *nagariensis*, *regA* is expressed exclusively in somatic progenitor cells, with its transcription beginning early in development shortly after inversion [49,50,51,52]. *regA* transcript levels appear to persist and fluctuate throughout the life cycle [49], but see the study by König and Nedelcu [24] for an alternative possibility and discussion.

The functional role of RegA, its amino acid composition, and the presence of a DNA-binding SAND domain in the RegA protein [53] helped establish the current working model in which RegA acts as a transcriptional repressor of genes needed for gonidial development [37]. A long-standing hypothesis is that *regA* suppresses the expression of nuclear-encoded chloroplast proteins required for chloroplast biogenesis and turn-over [54,55,56]. These negative effects on the chloroplasts would be reflected in the inability of the somatic cells to photosynthesize, grow, and divide. However, Matt and Umen [52] cast some doubt on this idea. They used whole transcriptome analysis to compare the expression profiles of germ cells and somatic cells. While photosynthetic genes were expressed at around two-fold higher levels in germ cells, photosynthetic genes were nevertheless highly abundant in somatic cells as well. Matt and Umen [52] propose that both germ cells and somatic cells maintain active photosynthesis, but germ cells are specialized in anabolic processes such as starch, fatty acid, and amino acid biosynthesis, while somatic cells break down starch and lipids to provide the substrates needed to synthesize ECM glycoproteins. Therefore, while it remains plausible that *regA* downregulates photosynthetic genes, it is also possible that *regA* downregulates other genes related to germ cell growth such as starch synthesis.

The structure of the *regA* gene has been well described for *V. carteri* and serves as the basic template for the gene structures of many other homologs of *regA* in the VARL (volvocine algae *regA-*like) gene family. The minimal promoter of *regA* consists of only 42 nt found directly upstream of the transcription start site with a plausible TATA box with the sequence TAATTGA beginning at −28 and an initiator region with the sequence CACTCAT beginning -1 relative to the transcription start site [57]. The transcriptional unit of *regA* is 12,477 nt long and contains 7 introns and 8 exons. After the introns are spliced out, the mature *regA* mRNA is 6725 nt long and consists of a 940 nt 5′UTR (exons 1–5), a 3147 nt coding region (exons 5–8), and a 2638 nt 3′UTR with a UGUAA polyadenylation signal [49] (Figure 3).

However, a splice variant that retains intron 7 (1194 bp) is expressed at low levels in *V. carteri* f. *nagariensis* as well. The donor splice site of intron 7 is GC instead of the typical GU, which may explain the variation in splicing. Remarkably, intron 7 encodes an ORF in the same frame as the rest of the *regA* coding region and, therefore, is likely to be translated, resulting in two different RegA protein products. However, experiments using modified *regA* transformation constructs to alter the splicing and translation of intron 7 have demonstrated that the presence or absence of intron 7 splicing has no detectable effect on the phenotypic rescue of regenerator mutants, despite the retention of intron 7 adding nearly 400 more amino acid residues to the RegA protein [57]. Interestingly, the homologous region to intron 7 is not spliced out in the closely related *V. carteri* f. *kawasakiensis*, and protein-level homology has been described in the intron 7 region across a wide variety of volvocine algae species [25,53]. Thus, it appears likely that splicing out intron 7 is a quirk specific to *V. carteri* f. *nagariensis*, while homologous regions are exonic in other species.

In addition to the promoter, the differential transcription of *regA* is regulated by two enhancers found in introns 3 and 5 and a silencer found in intron 7 [57]. Eight possible AUG start codons are found in the 5′UTR of mature *regA* mRNA and are thought to be bypassed via a ribosome shunting mechanism so that translation begins at the ninth AUG sequence of the mRNA [58].

Following translation, the predicted RegA protein is 1049-amino-acids-long without the inclusion of intron 7 or 1447 with intron 7 and contains a high proportion of glutamine, alanine, and proline residues [49,57]. A key structural region within the RegA protein is the VARL domain, which is the distinguishing feature of the VARL gene family [53,59] (Figure 4). The VARL domain is located between amino acids 444 and 558 in the RegA of *V. carteri* f. *nagariensis* and is composed of a highly conserved core VARL region (sites 484–558), a short but highly conserved N-terminal extension region (sites 444–455), and a less conserved linker region between these two [25,53,59]. In addition, two short motifs of high amino acid conservation have been identified that are shared across the predicted RegA proteins of numerous volvocine algae species: a “LALRP” motif upstream of the VARL domain and an “FLQ” motif found within the intron 7 region downstream of the VARL domain [25] (Figure 5).

The core VARL domain appears to encode a DNA-binding SAND domain [53]. The SAND domain (IPR000770/PF01342)—named after Sp100, AIRE-1, NucP41/75, and DEAF-1—is a DNA-binding domain found in animal and plant proteins that function in chromatin-dependent transcriptional control or bind-specific DNA sequences (e.g., [60]). SAND-containing proteins are involved in multiple distinct processes, both general and lineage/tissue-specific. However, most of the SAND-containing proteins with known functions are involved in multicellular development, including cell differentiation, cell proliferation, tissue homeostasis, and organ formation. For instance, DEAF-1 (Deformed Epidermal Autoregulatory Factor-1) is involved in breast epithelial cell differentiation in mammals [61] and is necessary for embryonic development in *Drosophila melanogaster* [62]. GMEB (Glucocorticoid Modulatory Element Binding) regulates neural apoptosis in the nematode *Caenorhabditis elegans* [63]. Spe44 (Speckled protein 44 kDa) is a master switch for germ cell fate in *C. elegans* and, like the mammalian AIRE1 (Autoimmune Regulator 1), plays a role in sperm cell differentiation [64,65,66]. In land plants, SAND domains are associated with ATX (the *Arabidopsis* homolog of trithorax) and ULTRAPETALA (ULT) proteins, which are involved in cell proliferation, cell differentiation, and tissue patterning. Specifically, ATX1 in *Arabidopsis thaliana* is required for root, leaf, and floral development through its histone methyltransferase activity [67], and ULT is a negative regulator that influences shoot and floral meristem size by controlling cell accumulation [68,69,70].

## 4. Evolution of the VARL Gene Family

The VARL gene family is defined by the presence of a homologous VARL domain within the predicted protein (note that volvocine algae possess additional SAND-containing proteins outside the VARL family; see next section and Figure 6). Although all VARL genes contain the VARL domain, the sequence level conservation outside of the VARL domain is very low. Thus, entire gene sequences cannot be aligned and used for phylogenetic analyses. The VARL domain itself is very short (~86 amino acids) and not highly conserved, such that its utility for inferring evolutionary relationships between the members of the VARL gene family is also limited. Nevertheless, information from gene synteny, sequence signatures outside of the VARL domain, and the locations of conserved introns can help draw more robust conclusions regarding the evolution of the VARL family. We summarize the available data below but direct readers to the study by Grochau-Wright et al. [25] (in particular, Supplementary Table S3 of [25]) for more detailed information.

Based on currently available whole genome sequence data, the VARL gene family contains 12 members in *C. reinhardtii* [59], 8 in *G. pectorale* [32] and *T. socialis* [72], 6 in *A. gubernaculifera* [33], and 14 in *V. carteri* [59]. With the exception of *regA* orthologs (when present), all other *regA* homologs are known as *regA*-like sequences, annotated as *RLS1-12* in *Chlamydomonas* and Goniaceae or *rlsA-O* in Volvocaceae.

To date, *regA* orthologs have been found in every member of the Volvocaceae family that has been investigated (including species without somatic cells) but appear to be absent in volvocine species outside of the Volvocaceae [25,32,33,41,59,72,73]. In all species with a *regA* ortholog for which a complete genome sequence is available, the gene is found in a syntenic gene cluster of 4–5 paralogs of closely related VARL genes called the *regA* cluster (Figure 2 and Figure 5). The first gene in the *regA* gene cluster is the *rlsA* gene, which has a unique highly conserved ~forty-amino-acid protein motif upstream of the VARL domain called “Pandorina’s Box” and a second short, conserved motif, named the “PRL” motif after its conserved sequence, downstream of the VARL domain [25] (Figure 5). Downstream of *rlsA* is *regA* followed by *rlsB*. Then, some species (i.e., *P. morum*, *P. caudata*, and *Y. unicocca*) have an additional paralog called *rlsO*. It appears that *rlsO* is found only in species outside of the “Eudorina-group” of the Volvocaceae, but further investigation is needed to confirm this hypothesis. These three *regA* cluster genes (*regA*, *rlsB*, and *rlsO*) all contain two short, conserved regions called the “LALRP” and “FLQ” motifs found upstream and downstream of the VARL domain, respectively. However, instead of *rlsO*, *V. ferrisii* has a different gene in the same location called *rlsN,* which is unique among all VARL genes because it has two VARL domains instead of one [73]. The relationship between *rlsO* and *rlsN* is not clear, though it seems plausible that *rlsO* underwent a domain duplication event to give rise to *rlsN* in *V. ferrisii*. Finally, the last VARL gene in the *regA* cluster is *rlsC,* whose orthologs do not appear to share any strongly conserved regions outside the VARL domain [25].

*C. reinhardtii* and other volvocine algae outside the Volvocaceae lack orthologs of any of the *regA* cluster genes. The closest homolog to the *regA* cluster genes found in these species is *RLS1*. This gene is an ortholog of the Volvocaceaen *rlsD,* which is the closest *rls* paralog of the *regA* cluster. Duncan et al. [59] proposed that in the common ancestor of the lineages leading to *V. carteri* and *C. reinhardtii,* a VARL gene underwent duplication to give rise to two paralogs they referred to as “proto-*RLS1/rlsD*” and “proto-*regA*”. Following the separation of the *C. reinhardtii* and *V. carteri* lineages, they suggested that proto-*regA* was lost from *C. reinhardtii* but underwent additional duplication in *V. carteri’s* lineage to give rise to the *regA* cluster. Meanwhile, proto-*RLS1/rlsD* was retained in both lineages and evolved into the modern-day *RLS1* and *rlsD*. However, the addition of more *regA* sequences from a variety of species and lack of a proto-*regA* candidate in *G. pectorale*, *T. socialis*, and *A. gubernaculifera* favor a different model in which the ancestral *RLS1/rlsD* underwent a series of tandem duplications to form the *regA* gene cluster at the origin of the Volvocaceae [25,32].

Finally, several lineage-specific details about the *regA* cluster and *rlsD* are relevant to further understanding the evolution of this gene family in different volvocine species. First, in all species for which sufficient data are available, the *rlsD* gene is found near the *regA* gene cluster, except for *V. carteri* (Figure 2). This supports the idea that *RLS1/rlsD* duplicated to give rise to the *regA* cluster; however, later, in the *V. carteri* lineage, *rlsD* translocated away from the *regA* cluster [25,59]. Second, the *regA* cluster is inverted with respect to nearby syntenic markers in *P. caudata*, although the *regA* cluster genes themselves are in the same order relative to each other. Third, the *regA*, *rlsB*, and *rlsO* genes in *Y. unicocca* are more similar to each other than they are to their homologs in other species, suggesting two of these genes were lost and were later replaced via the duplication of the remaining gene (thus, these three genes in *Y. unicocca* are not orthologs of the similarly labeled genes in other species) [25]. It is possible that similar mechanisms of complex evolution have happened within the *regA* cluster in other species as well, but phylogenomic analysis supports the orthology of *regA* cluster genes within the Eudorina group species [41].

Synthesizing the information above, we propose the following scenario for the evolution of the *regA* gene cluster (Figure 2). The VARL gene family comprising several paralogs including *RLS1/rlsD* was already present in the common ancestor of all volvocine green algae. *RLS1/rlsD* underwent one or more duplication events in the common ancestor of the Volvocaceae family to give rise to a five-gene *regA* gene cluster comprising *rlsA, regA, rlsB, rlsO*, and *rlsC*. After the lineage leading to *V. ferrisii* diverged from the rest of the Volvocaceae, its *rlsO* gene gained a second VARL domain and evolved into *rlsN*. Meanwhile, the common ancestor of the Eudorina group lost *rlsO*. In addition, *Y. unicocca* lost two internal *regA* cluster genes (*regA*, *rlsB*, or *rlsO*) but restored the five-gene cluster via gene duplication, and the *regA* cluster of *P. caudata* became inverted relative to nearby syntenic markers (Figure 2).

## 5. SAND-Domain-Containing Sequences beyond Volvocine Algae 

The VARL domain is postulated to be specific to the volvocine VARL gene family but contains a SAND domain similar to that seen in plants and animals. The evolutionary origins of the VARL family and its relationships with other SAND-domain-containing proteins are unclear [71]. Interestingly, SAND-domain-containing proteins appear to be restricted to Viridiplantae (green algae and land plants, also known as green plants) and Metazoa. Specifically, SAND-containing sequences were detected in both lineages within Viridiplantae: Streptophyta (land plants and their closest green algal relatives, the Charophytes) and Chlorophyta (green algae in the Chlorophyceae, Trebouxiohyceae, Ulvophyceae, and Prasinophytes, a paraphyletic group of early-diverged single-celled lineages). However, despite the presence of SAND-containing proteins in all metazoan lineages (from sponges, ctenophores, and cnidarians to mammals), SAND sequences could not be detected in their closest unicellular relatives (choanoflagellates, filasterians, and ichthyosporeans) [71].

In SAND-containing proteins, the SAND domain is found either alone (as in the VARL family) or in various combinations with one or more domain types (Figure 6). Notably, no multi-domain architecture is shared between the SAND-containing animal and green plant proteins. Furthermore, the range and distribution of architectures are very different between green plants and animals. For instance, a rich toolkit of SAND-containing proteins with various (and complex) architectures is present in green algae (including the volvocine species), but only two architecture types are found in land plants. On the other hand, only one or two architectures are found in sponges and cnidarians, but a richer repertoire (six or more types) is present in vertebrates (Figure 6).

Moreover, phylogenetic analyses did not reveal any orthologous relationships between SAND-domain-containing sequences in the animal and green plant lineages [71]. In addition, this limited distribution of SAND sequences is intriguing. Such a “patchy” distribution (Figure 7) is considered to be indicative of lateral gene transfer (LGT) [74], and this was also suggested to be the case for SAND [71]. Phylogenetic analyses and the presence of a specific sequence motif suggest that animal SAND-containing proteins evolved via an LGT event from a VARL-like sequence [71].

Metazoans, streptophytes, and volvocine algae evolved multicellular development independently ca. 600 MYA, >700 MYA, and 200 MYA, respectively [22,76]. The fact that they all employ SAND-containing proteins in processes involving the regulation of cell proliferation and differentiation suggests that SAND-containing proteins were co-opted for and deployed in similar developmental processes in parallel. Thus, it has been suggested that the independent evolution of complex development in these lineages involved the parallel deployment of ancestral sequences containing a SAND domain [71]. Furthermore, the presence of SAND-containing proteins in single-celled green algae (Figure 6) argues that the ancestral role of this domain was in the regulation of gene expression outside a multicellular context. Below, we review the role of such a SAND-domain-containing sequence in a single-celled volvocine species and its co-option for the regulation of cell proliferation and somatic cell differentiation during the evolution of the volvocine *regA*-like family.

## 6. Functional Evolution of *regA* via Co-Option of a Life History Trade-Off Gene

*regA* has been known to act as a master regulator of somatic cell differentiation in *V. carteri* for over two decades [49]. However, it is still not known how *regA* is differentially regulated and how exactly it acts to suppress division in cells that fall under the 8 µm threshold size at the end of embryogenesis. Evolutionary approaches can provide alternative or additional means to gain insight into the function of a gene. Based on its role in suppressing reproduction in somatic cells, it has been hypothesized that *regA* evolved from a gene that was involved in trading off reproduction for survival (i.e., a life history trade-off gene) in the single-celled ancestors of *V. carteri*. Specifically, such a gene could have been co-opted by changing its expression from a temporal context (in response to an environmental cue) into a spatial context (in response to a developmental cue) [17] (Figure 8).

### 6.1. Life History Trade-Offs in Single-Celled Organisms 

Although single-celled organisms, by definition, do not possess specialized cell types, they are nevertheless capable of differentiating themselves into various cell states in response to environmental changes. These capabilities include simply switching between the activities enhancing survival, growth, or reproduction during their lifetime, as well as adopting more stable and functionally distinct states (e.g., gametes or spores). The need to switch between survival and reproduction to increase overall fitness reflects one of the main trade-offs that characterize the life history of all organisms [77,78,79]. The mechanistic basis of life history trade-offs can be resource availability, structural constraints, cellular processes or genes that affect two sets of activities in opposite ways, or a combination of such factors [77,78,80,81,82,83,84,85,86,87,88]. Life history trade-offs can be amplified during certain environmental conditions, such as nutrient limitation, in which survival is prioritized over reproduction [87,88].

In photosynthetic organisms, including volvocine algae, nutrient limitation induces a series of physiological changes—known as acclimation—characterized by the downregulation of photosynthesis to avoid oxidative damage associated with imbalances between light excitation and NADPH consumption [89]. In single-celled volvocine algae, such acclimation processes that increase survival also result in a temporary cessation of growth (dependent on photosynthesis) and reproduction (dependent on growth) [90].

### 6.2. Chlamydomonas RLS1 Is a Life History Trade-Off Gene Induced via Environmental Cues 

As noted earlier, the closest homolog of *regA* in *C. reinhardtii* is *RLS1* [17,59]. Consistent with the proposed hypothesis that *regA* evolved from a life history trade-off gene, initial studies showed that *RLS1* expression is upregulated under nutrient (phosphorous or sulfur) and light deprivation, which requires the suppression of reproduction to increase survival [16,17]. Later, to confirm that *RLS1* acts as a bona fide life history trade-off gene, the reproduction and survival of an RLS1 mutant during phosphate deprivation were investigated [91]. As expected, the *RLS1* mutant was not able to suppress its reproduction when phosphate was limited. In fact, the population size of the mutant exceeded that of the wild-type. However, this short-term immediate reproductive advantage was counteracted by a loss in long-term viability, arguing that *RLS1* is a genuine life history trade-off gene [91].

Theoretically, the suppression of reproduction in nutrient-limiting conditions can involve three distinct possibilities [91]. First, the suppression of reproduction and increased survival could be a direct response to the reallocation of nutrients, possibly involving a trade-off between protein biosynthesis (growth) and energy metabolism (survival) [77]. Alternatively, reproduction could be suppressed in response to nutrient-stress-induced production of reactive oxygen species (ROS); such oxidative stress could trigger temporary cell cycle arrest (suppression of reproduction) to repair the oxidative DNA damage. Lastly, the suppression of reproduction could be induced by redox signals associated with imbalances between excitation energy and electron acceptor levels under nutrient deprivation [92]. Such signals trigger the downregulation of photosynthesis to avoid potential photo-oxidative damage, which will increase survival but also limit cell growth and thus reproduction. The available data are consistent with the latter scenario [91]. Thus, mechanistically, *RLS1* appears to be induced by a redox imbalance/signal, and once expressed, RLS1 could act via the downregulation of photosynthesis. Consistent with this model, the induction of *RLS1* coincides with the downregulation of a light-harvesting chloroplast protein-coding gene, and the experimental inhibition of photosynthetic electron transport can induce the expression of *RLS1* [16].

### 6.3. V. carteri regA Retained the Ancestral Environmental Regulation

Nonetheless, how did the regulation of *regA* in somatic cells evolve from the environmentally induced regulation of its *RLS1*-like progenitor? To address this question, König and Nedelcu [24] proposed two scenarios (Figure 9): (i) a new developmental regulation replaced the ancestral environmental regulation of *RLS1/rlsD* in the *regA* paralog, or (ii) a new developmental regulation was added to the ancestral regulation. The second scenario predicted that in addition to its developmental expression, *regA* could also be induced in response to environmental cues. Recently, it was shown that *regA* can, indeed, be expressed in both developmental and environmental contexts (see scenario ii in Figure 9A). Specifically, *regA* was induced in response to light exposure following an extended dark period in a *V. carteri* mutant that lacks cell differentiation and a functional RegA protein but still expresses *regA* developmentally [24]. Furthermore, because the expression of *regA* was affected by the duration of both dark and light exposure, it is likely that the environmental induction is triggered by a metabolic imbalance [24].

### 6.4. Co-Option of an Environmentally Regulated Gene into a Developmental Master Regulator 

To address how the new developmental regulation evolved, two models have been proposed: a completely new signaling pathway evolved, or the ancestral signaling pathway was co-opted into a developmental context [24]. The two models make different predictions as to the regulation of *regA* in *V. carteri* (Figure 9B).

The first model implies that new *cis*-regulatory and/or *trans*-acting elements were added to the ancestral *RLS1/rlsD* gene regulation. Although the developmental regulation of *regA* in *V. carteri* is known to involve intronic *cis*-regulatory elements [57], their sequence as well as the *trans*-acting factors binding to them are still unknown. Moreover, nothing yet is known about the regulatory sequences of *RLS1*. Notably, *RLS1* does not share *regA*’s exon–intron structure and/or similar intronic sequences; thus, there is no direct correspondence between the postulated intronic silencers and enhancers in *regA* and potential regulatory elements in *RLS1* (Figure 3).

Changes in the deployment of *trans*-acting factors, *cis*-regulatory elements (de novo or via the modification of pre-existing elements), or a combination of both have been proposed to have taken place during the evolution of morphological innovations in animals [93]. Similarly, the acquisition of new, distal promoters was invoked in the co-option of ancestral cAMP signaling genes for new developmental roles in the social amoebae *Dictyostelium discoideum* [94]. Finding a similar mechanism in volvocine algae would argue for its general role in the changes in gene regulation during the evolution of major morphological innovations. Genes with dual regulation are often seen as intermediate steps during the sub-functionalization process that results in two specialized genes (e.g., [95]). In this context, it is relevant to know if *regA*’s paralogs (*rlsA*, *rlsB*, *rlsO*, and *rlsC*) have retained the ancestral regulation or whether they specialized into specific developmental roles. A preliminary investigation found that the genes *rlsB* and *rlsC* are co-expressed with *regA* during development, but the functional significance of this is not yet known [50]. Likewise, a preliminary investigation using RNAi to knock down *rlsA, rlsB,* and *rlsC* expression in *V. carteri* did not result in any discernable vegetative phenotype [96]. The mechanisms underlying these results and their implications are not yet clear and require further investigation. Notably, of the total 14 VARL genes (including *regA*) in *V. carteri*, 10 are overexpressed in somatic cells, 3 are expressed constitutively, and 1 is overexpressed in reproductive cells [51].

The second model requires that the same intracellular signal that induces the environmental expression of *RLS1* in *C. reinhardtii* is also triggered in *V. carteri*’s small cells at the end of embryogenesis. As *RLS1* is likely part of the general acclimation response [16], it is possible that its induction is the result of an energetic imbalance mediated via a redox signal (e.g., NADPH/NADP+ and ROS) (e.g., [97]). A similar signal could be responsible for the observed induction of *regA* in *V. carteri* cells exposed to light after long dark periods [24]. If the same signal is also produced in small cells at the end of embryogenesis (induced by a different imbalance caused by small cell size), *regA* could be expressed in a developmental context using the same environmental regulatory elements. The postulated developmental signal can be triggered by an imbalance between membrane-bound proteins (e.g., electron transport carriers and ion transporters) and soluble factors (e.g., NADP+) as the surface-to-volume ratio in these small cells is in favor of membrane proteins (see Figure 2 in [16]). Interestingly, the environmental induction of *regA* is also affected by cell size [24]. If this model is correct, finding the exact signal involved in the environmental induction of *regA* could help answer the long-standing question of how small cell size determines somatic cell differentiation in *V. carteri*.

Notably, the conditions that induced *regA* in regA mutant cells ultimately resulted in the activation of programmed cell death (PCD), while wild-type somatic cells (harboring a functional RegA protein) show little response in terms of environmental *regA* induction or loss in viability [24]. Furthermore, wild-type somatic cells are unaffected by heat stress, while regA mutant cells as well as wild-type gonidia (which do not express *regA* [49]) undergo PCD [98,99]. Altogether, these findings suggest that in addition to its role in suppressing the reproduction of somatic cells, the presence of a functional RegA protein in somatic cells confers (directly or indirectly) resistance to environmental stress.

The fact that somatic cells are already protected (see [24] for a discussion of this) by the presence of the developmentally expressed RegA protein raises the question of why the environmental regulation of *regA* is still maintained in *V. carteri*. It has been proposed that, similar to *RLS1* in *C. reinhardtii*, *regA* plays a direct role in the response to stress in gonidia [24]. For instance, under nutrient deprivation, gonidia stop growing and undergo a temporary cessation of reproduction. At the cell level, the inhibition of gonidial growth and reproduction can be an acclimation response that prevents the accumulation of oxidative damage and thus ensures survival. At the multicellular level, this is an adaptive response that is costly in the terms of immediate reproduction but beneficial in terms of offspring quality since it avoids ROS-induced DNA damage and mutations in the gonidia. Nevertheless, when damage is extensive (such as during heat stress), PCD is the best adaptive response as it eliminates potentially damaged gonidia and/or prevents the transmission of deleterious mutations to offspring [99].

Preliminary investigations into the effects of knocking down or overexpressing *rlsD* during *V. carteri* development are also compelling [96,100]. Knocking down *rlsD* expression appears to result in colonies with reduced size or with germ cells that divide but fail to complete development. In contrast, the overexpression of *rlsD* results in the development of “somagonidia”, wherein germ cells do not grow to their normally large size but are still larger than soma and exhibit soma-like characteristics such as the presence of eyespots and flagella [96]. The gene expression profiles of wild-type *V. carteri* compared with those with *rlsD* overexpression show downregulation of ribosome- and photosynthesis-related genes [100]. Together, these results suggest that *rlsD* plays an important complementary role to *regA* in regulating cell growth, supporting the idea that the ancestral functions of *RLS1/rlsD* were sub-functionalized and/or neo-functionalized as *regA* evolved following its duplication from *RLS1/rlsD* [100].

## 7. The General Role of Stress and Life History Trade-Off Genes in the Re-Organization of Fitness during the Evolution of Multicellularity

Stress responses take on special significance during evolutionary transitions in individuality more generally. As discussed in the Introduction, during an evolutionary transition in which individuals form groups that evolve into a new kind of individual, fitness must be reorganized so that it becomes a property of the group. The fitness opportunities of the previous individual must be significantly reduced or eliminated. The readjustment of fitness components during stressful periods provides a useful substrate for the reorganization of fitness because mechanisms are already in a position within the previous lower-level individual to adjust fitness components during times of stress [12,16,17,24].

The findings that *regA* is a master developmental regulator that (i) evolved from a stress-induced gene, (ii) still manifests its ancestral environmental regulation, and (iii) confers stress protection offers a direct link between stress responses and the early evolution of somatic cell differentiation. In stressful environments, fitness is compromised, and fitness components must be adjusted to meet the challenge if the individual is to survive. In particular, survival will often be prioritized over reproduction in the short term. We suggest that the ability to re-organize fitness components under stress can be co-opted during the evolution of multicellularity and can contribute to the evolutionary potential and stability of multicellular lineages.

In *V. carteri*, the co-option of a life history trade-off gene that could suppress reproduction at the cell level increased the survival of the multicellular individual, since non-dividing somatic cells could maintain flagellar motility throughout the life cycle. However, as *regA* evolved from a life history trade-off gene expressed under stress, the permanent suppression of reproduction also provided somatic cells with survival cell-level benefits in terms of enhanced resistance to environmental stress. Indeed, the lack of a functional RegA protein (such as in *regA* mutants) makes cells more sensitive to environmental stress [98]. This increased sensitivity to stress can also contribute to the stability of the multicellular individual as *regA* mutant cells that regain reproductive capabilities (and negatively affect the fitness of the multicellular individual) will incur a cost in terms of survival [98]. On the other hand, the developmental repression of *regA* expression in gonidia is required to allow the reproduction of the individual. However, since *regA* has maintained its ancestral environmental regulation, gonidia can adaptively respond to environmental stress by inducing either temporary cell cycle arrest (to prevent or repair damage) or PCD (to prevent the transmission of deleterious mutations to offspring).

As life history trade-offs are common in single-celled organisms (e.g., [101,102,103,104]), it has been suggested that similar co-options of life history trade-off genes with antagonistic effects on survival and reproduction have taken place during the evolution of multicellularity in other lineages [98]. Notably, the de-differentiation and increased proliferation of cancer cells were also shown to result in increased sensitivity to nutrient stress due to their failure to trade off cell proliferation for maintenance in stressful environments [105,106]. This trade-off, likely inherited from the unicellular ancestors of animals, might have contributed to the stability of the multicellular individuals during early animal evolution and might still be involved in the purging of most pre-cancerous cells [98]. Recently, pre-existing ancestral stress responses have been linked to the evolution of several other developmental processes, from aggregative multicellularity in *D. discoideum* to the differentiation of decidual stromal cells in placental mammals [107,108,109]. 

## 8. Future Directions

Despite the unprecedented suitability of the volvocine algae and the *regA*-like family for understanding the mechanistic and genetic basis for the transition to a new higher-level individual through the reorganization of fitness components, there are many unknowns yet to be addressed. For instance, while the role *regA* plays in cellular differentiation is clear in *V. carteri* f. *nagariensis,* the specific details of how it carries out this function are not well understood. The expression of *regA* is regulated by cell size at the end of cleavage in *V. carteri* [39], but how cell size induces the expression of *regA* is unknown. Similarly, how *regA* regulates its target genes and what those target genes are is unresolved.

The function of *regA* is even less clear in other volvocine algae species. Cellular differentiation arose three to five times in the Volvocaceae [25,27,35] (Figure 2), leaving open the question of whether *regA* was co-opted to control differentiation multiple times or if different genes control differentiation in other volvocine species with somatic cells. Intriguingly, *regA* is found in species lacking somatic cells, but its function in these undifferentiated species is not known [25]. Some volvocine algae thought to lack differentiation, such as *Eudorina*, can develop somatic-like cells under environmental stress [110]. Whether or not *regA* is involved in this phenomenon is currently unknown as well. A better understanding of how *RLS1* acts and is regulated in *Chlamydomonas* and volvocine species that lack cellular differentiation will help fill in the gaps of how *regA* was co-opted to control somatic cell differentiation as well. For instance, the *cis*-regulatory elements of *RLS1* are undescribed. Identifying these regulatory elements and comparing them to the *cis*-regulatory elements of *V. carteri regA* would illuminate how the environmentally induced regulation of *RLS1* was co-opted during the evolution of somatic cell differentiation.

Understanding the functions of the *regA* cluster genes other than *regA* also requires more work. Preliminary studies have shown that at least *rlsB* and *rlsC* have a similar expression pattern as *regA* during *V. carteri* development [50], which suggests that the *regA* cluster as a whole may be involved in development and differentiation, but more work needs to be undertaken to fully assess this idea [96]. Interestingly, Grochau-Wright et al. [41] found that a morphological mutant of *V. powersii* that has fewer cells and a lower soma-to-germ ratio than the WT strain has a mutation in its *rlsB* gene. Nevertheless, the transformation of the WT-*rlsB* gene into this mutant did not lead to morphological rescue. Thus, the cause of the mutant’s altered phenotype and the functional significance of the *rlsB* mutation (if any) remain unclear.

Very little is known about the structure, function, and evolution of VARL genes outside of the *reg* cluster due to less attention and less effort being put into cloning and sequencing these genes. However, whole genome sequence data are available for several species that require further analysis and annotation to determine the total number of VARL genes present, specifically *E. elegans*, *Y. unicocca*, *Volvox reticuliferus*, and *Volvox africanus* [111,112]. In addition, Lindsey et al. [35] generated a large transcriptomic dataset for 47 volvocine algae species that could potentially be mined for VARL gene family members. Searching these already available data for VARL genes could substantially expand our knowledge of the structure and evolution of VARL genes outside of the *reg* sub-family. Intriguingly, Klein et al. [51] found that the *rlsM* gene in *V. carteri* is overexpressed in reproductive cells and suggest a possible role of this VARL gene in germ cell development.

In addition, understanding how species in the genus *Astrephomene* control their cellular differentiation is of interest. The *Astrephomene* species are the only volvocine algae outside of the Volvocaceae that have cellular differentiation, but unlike the *Volvox* and *Pleodorina* species, the somatic cells of *Astrephomene* are located in the posterior of the colony due to differences in colony development [113]. In addition, *Astrephomene* does not possess the *regA* gene and evolved somatic cells independently of other volvocine algae [25,33,35] (Figure 2).

A key lesson from studying *RLS1* and *regA* in the volvocine algae is that life history genes may be co-opted for the cell differentiation of soma, in particular, environmentally controlled growth suppression genes can be co-opted to provide the genetic basis for the developmental control of cell differentiation. The need for single-celled organisms to respond to times of stress by decreasing their growth to support their survival provides the foundation for the evolution of somatic cells in a multicellular organism [16,17,19]. *regA* is the best-known example of such a life history gene; however, as *regA* is specific to the volvocine algae clade, it would be useful to know if genes with similar life history functions were co-opted to produce soma in other clades that made the transition to multicellularity independently. Similarly, the evolution of specialized reproductive and worker castes during the transition to eusociality is analogous to the evolution of germ and soma during the transition to multicellularity. Thus, it may be fruitful to investigate the co-option of stress response life history genes during the transition to eusociality as well.

## 9. Summary

The evolution of cellular differentiation is a key event during the transition from single-celled to multicellular life. Specialized germ cells and somatic cells reorganize the two essential components of fitness between different cell types, thereby transferring fitness from the cell level to the multicellular level. Understanding the genetic basis for the evolution of cellular differentiation during unicellular-to-multicellular transitions is a major challenge in evolutionary biology. The co-option of life history trade-off genes present in unicellular organisms that differentially affect survival and reproductive functions in response to the environment is one route for the evolution of genes controlling cellular differentiation. The *regA*-like gene family of the volvocine green algae is an unrivaled model system to study this co-option due to the recent origin of multicellularity in this clade and the presence of extant relatives at different levels of multicellular complexity and individuality.

The *regA* gene in *V. carteri* f. *nagariensis* is the type-gene for the VARL gene family specific to the volvocine algae lineage. The common ancestor of *V. carteri* and *C. reinhardtii* likely had several VARL gene family members, one of which was *RLS1*. The *RLS1* gene duplicated several times to give rise to the *regA* gene cluster in the common ancestor of the Volvocaceae, setting the stage for the functional co-option of *regA* during the evolution of cellular differentiation as well as other lineage-specific changes to *regA* cluster genes (Figure 2). The co-option of *RLS1′s* functions into a *regA*-like gene responsible for somatic cell differentiation likely involved the simulation of the ancestral environmentally induced signal in a developmental context (Figure 9).

The defining feature of all VARL genes is the presence of the VARL domain, which contains a conserved SAND domain. The SAND domain is found in other green algae, land plants, and animals but appears to be missing from other eukaryotic lineages. This indicates the possibility that the SAND domain was horizontally transferred between green algae and animals early on in eukaryotic evolution, wherein it appears to have been co-opted multiple times independently in a variety of developmentally important functions.

Overall, we argue that the co-option of life history trade-off genes during the transition to multicellularity underlies the re-organization of fitness between soma and germ to optimize fitness at the multicellular level and enhance the individuality of the multicellular group. It is axiomatic that all organisms must have such stress response trade-off genes, and it remains to be determined whether or how fitness reorganization and the co-option of life history genes during the evolution of specialized cell types apply to other lineages that evolved multicellularity independently.

## Figures and Tables

**Figure 1 genes-14-00941-f001:**
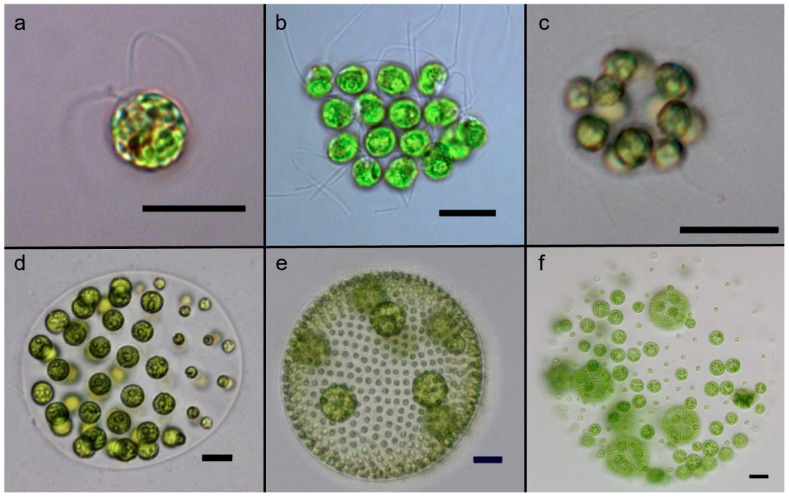
Select species illustrating volvocine green algae diversity in complexity and individuality. (**a**) *Chlamydomonas reinhardtii* (scale bar 10 µm), (**b**) *Gonium pectorale* (scale bar 10 µm), (**c**) *Eudorina elegans* UTEX 1212 (scale bar 10 µm), (**d**) *Pleodorina californica* (scale bar 25 µm), (**e**) *V. carteri* f. *nagariensis* (scale bar 50 µm), and (**f**) *V. carteri* regenerator mutant showing dedifferentiated somatic cells (scale bar 50 µm). Images (**a**–**e**) reproduced from the study by Grochau-Wright et al. [25].

**Figure 2 genes-14-00941-f002:**
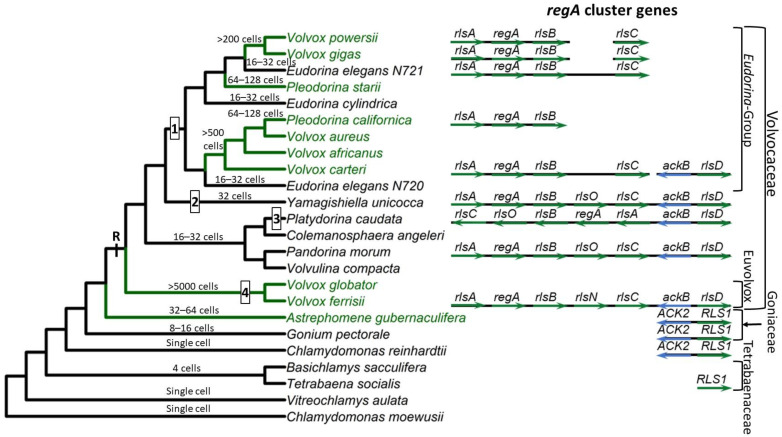
Origin and evolution of the *regA* gene cluster in volvocine algae. Phylogenetic tree shows relationships between selected volvocine algae species based on topology from the study by Lindsey et al. [35], with families and major clades of volvocine species indicated using brackets on the right. Species in green have obligate somatic cells, while species in black are undifferentiated. The numbers and positions of origins of somatic cells are consistent with the studies by Grochau-Wright et al. [25] and Lindsey et al. [35]. Typical cell numbers for specific species or lineages are indicated above the branches. Currently available *regA* gene cluster sequences and assemblies are shown to the right. *regA* and *rls* genes are shown in green, while nearby syntenic marker gene *ACK2/ackB* is shown in blue. Gene cluster diagrams show assembly status and completeness but are drawn to maintain alignment of homologs not to scale of actual genomic distances. Note that *P. californica* is assumed to possess an *rlsC* gene that has not yet been sequenced. Major events in the evolutionary history of the *regA* gene cluster marked on phylogeny: R = origin of *regA* gene cluster, 1 = loss of *rlsO*, 2 = loss and reduplication of *regA*, *rlsO*, and/or *rlsB* in *Y. unicocca*, 3 = inversion of *regA* cluster relative to nearby syntenic genes in *P. caudata*, and 4 = transformation of *rlsO* into *rlsN* through domain duplication.

**Figure 3 genes-14-00941-f003:**
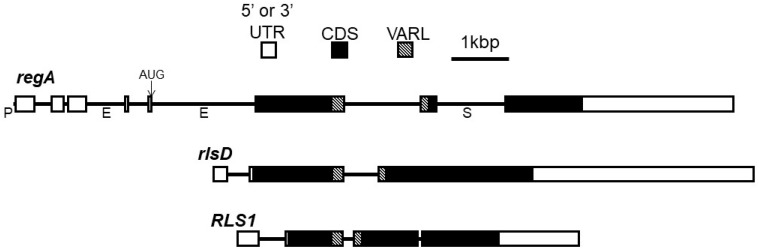
Gene structure diagrams for *regA* and *rlsD* in *V. carteri* f. *nagariensis* and *RLS1* in *C. reinhardtii*. Exons are shown as boxes with untranslated regions in white and coding regions in black. The portion of the coding region encoding the VARL domain is indicated with striped pattern. Introns are shown as lines between exons, and the 42 bp minimal promoter of *regA* is shown before first exon and labeled as “P”. *regA* introns labeled as “E” and “S” denote locations of enhancer and silencer *cis*-regulatory elements, respectively. Figure based on the study by Kirk [37].

**Figure 4 genes-14-00941-f004:**
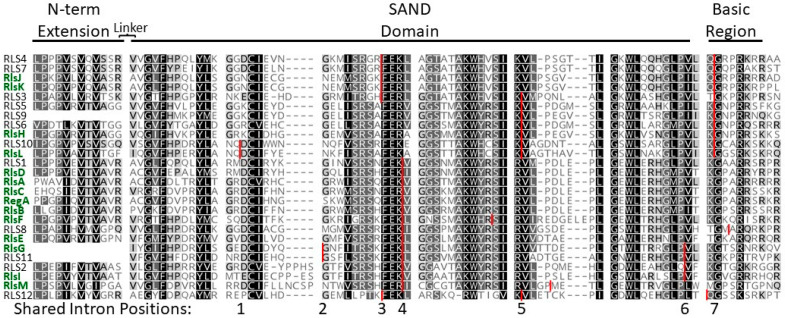
Alignment of VARL domains in *C. reinhardtii* and *V. carteri*. Names of *V. carteri* VARL domains are shown in bold green font. Conserved regions are annotated above alignment. Variable linker region between N-terminal extension and core VARL SAND domain not shown. Vertical red lines indicate conserved intron positions. Figure adapted from the paper by Duncan et al. [59].

**Figure 5 genes-14-00941-f005:**
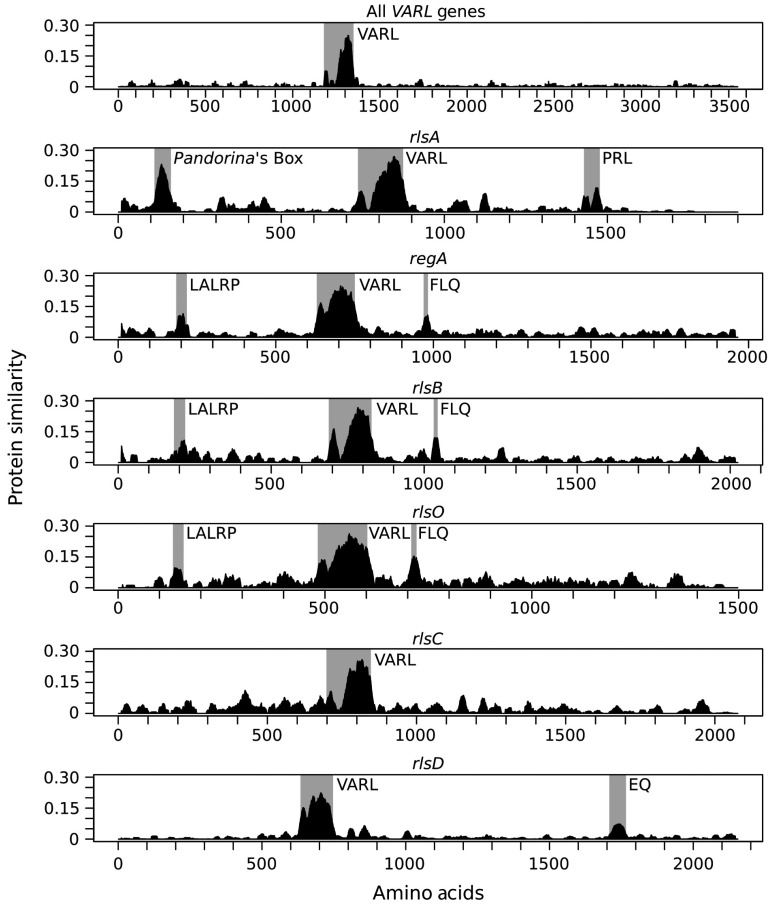
Protein similarity plots for all regA clusters and RLS1/rlsD proteins based on syntenic positions (rlsA, regA, rlsB, rlsO, rlsC, and rlsD). Regions showing high similarity are highlighted with grey boxes. The two peaks in the shaded VARL region represent the N-terminal extension and core VARL domain separated by the less conserved linker region. Reproduced from the study by Grochau-Wright et al. [25].

**Figure 6 genes-14-00941-f006:**
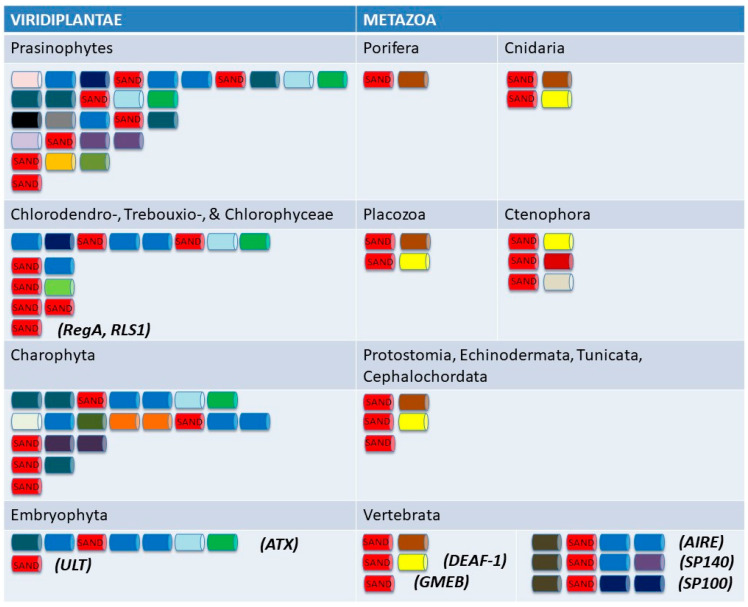
Most common domain architectures in SAND-domain-containing proteins present in the main Viridiplantae and Metazoa lineages showing differences in domain architectures between green alga/plant and animal SAND-containing proteins (various colors indicate the different domains found in specific proteins and lineages; for domain names see the study by Nedelcu [71]). Examples of proteins with known functions are indicated in parentheses. Adapted from the study by Nedelcu [71].

**Figure 7 genes-14-00941-f007:**
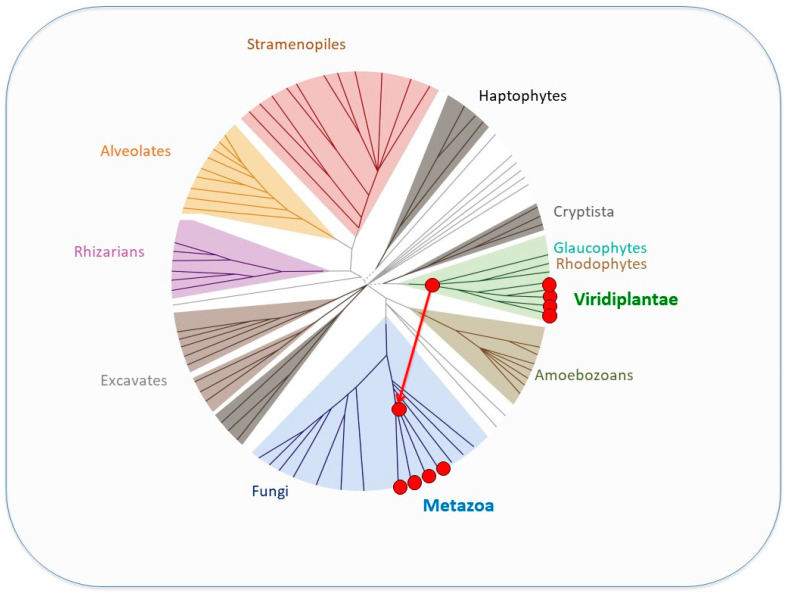
Unrooted eukaryotic tree (adapted from the study by Keeling & Burki [75]) showing the phylogenetic distribution of SAND sequences (red dots indicate occurrence and the hypothesized lateral gene transfer (red arrow) from a green algal ancestor to an early metazoan (adapted from [71]). For further discussion on the postulated lateral gene transfer, see the study by Nedelcu [71].

**Figure 8 genes-14-00941-f008:**
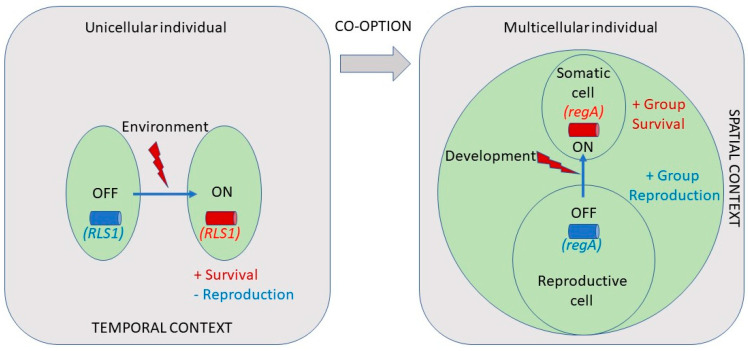
Co-option of a life history gene (e.g., *RLS1*) that traded off reproduction for survival in a single-celled ancestor into a regulator of soma/germ cell differentiation (e.g., *regA*) in a multicellular descendant by changing its expression from a temporal context (in response to an environmental cue) into a spatial context (in response to a developmental cue). The co-opted gene is inactive in reproductive germ cells but becomes active in somatic cells during development. Adapted from the study by Nedelcu and Michod [17].

**Figure 9 genes-14-00941-f009:**
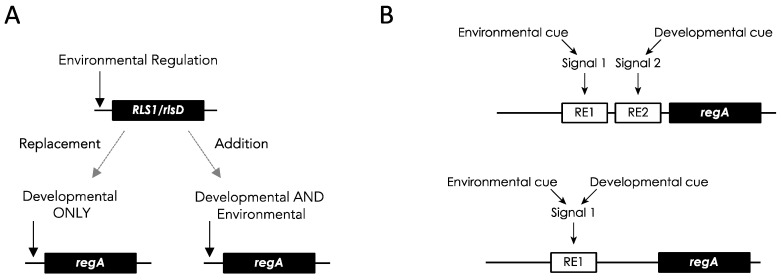
(**A**) Two potential scenarios envisioning the co-option of an ancestral environmentally regulated *RLS1*/*rlsD*-like gene into a developmentally regulated *regA*, involving either (i) the replacement of the ancestral environmental regulation or (ii) the addition of a new layer of regulation. (**B**) Two models to account for the observed dual regulation (environmental and developmental) of *regA* in *V. carteri* involving (i) the acquisition of a new signal transduction pathway and regulatory element (RE2) or (ii) the co-option of the environmentally induced intracellular signal and regulatory element (RE1). Adapted from König & Nedelcu [24].

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
