# Peer review of "The Genetics of Fitness Reorganization during the Transition to Multicellularity: The Volvocine *regA*-like Family as a Model"

_genes, 2023, doi:10.3390/genes14040941_

Round 1
Reviewer 1 Report
Grochau-Wright et al. discuss the origins of germ-soma differentiation in multicellular organisms and the strategic shift of survival-reproduction allocation, as exemplified by the regA protein and its relatives in the volvocine green algae that are a model system to infer the path from unicellular organisms to multicellular ones with germ-soma differentiation They depict that shift as "fitness reorganization." Specifically, the authors focused on the Volvox carteri germ-soma differentiation master regulator gene, or somatic regenerator (regA), the gene duplication events which produced the regA/VARL family, and their role in the common unicellular ancestor as a putative stress response transcription factor gene. Their idea on the co-optional shift of temporal stress response mechanism to spatial and developmental germ-soma dichotomy is interesting, and this review provides a nice deep dive into current ideas about how regA like genes may have evolved to govern the emergence of cell-type specialization. Some suggestions to improve the manuscript are below.
P.5
On Figure 2, multicellular traits (cell number, spheroidal/planar morphology, and presence of germ-soma) and the sub-families/clades in volvocine algae should be indicated/labeled as those matters are discussed in the main text.
P.6
This manuscript focuses on the development of V. carteri and the emergence of somatic cells. To discuss the emergence of multicellular development in the evolutionary context, however, it may be appropriate to discuss other aspects of multicelluarity in volvocine algae that are also integral to its evolution. For example, multiple fission, which already existed as a form of temporal cell-type specialization in the normal life cycle of unicellular ancestors, is a critical part of the life cycle that was modified to enable multicellular evolution. The role played by regA in V. carteri is critical for proper development, but many or most of the key traits that govern fitness in multicellular volvocines, including V. carteri, are not related to creating a dedicated somatic lineage, but instead to morphogenetic processes that allow a coherent multicellular individual to function (e.g. intracellular bridges, inversion, spheroid polarity, extracellular matrix, delayed cell division/extended flagellated stage etc.) Interestingly, even though regA is expressed early in somatic development, it is not strictly essential for soma to differentiate, but is required to stay differentiated. Moreover, it remains unclear how important are regA and VARL genes in other multicellular volvocines for development or cell type specialization. Overall, I feel that a more balanced discussion of these topics will provide better context for a general audience and would not detract from a focus on the very interesting evolution of the VARL family.
P.6, paragraph 2
Related to above, it would be worth mentioning other developmental genes identified in V. carteri that have homologs in C. reinhardtii or other volvocine genera such as glsA.
P. 6. Middle paragraph. “On the other hand, a set of lag genes are specifically induced in the large cells . .”The identity of lag genes is unknown, so how could we know if they are induced at a specific time in development?
PP.6-10
On regA and its paralogs, they refer to the gene structure and conserved regions in the VARL family. There is too much detailed discussion of these domains given the limited understanding of their importance.
P.8
On Figure 4, the numbers above the alignment is confusing. They are not based on locations in any of the proteins. In the same figure for "conserved intron positions" the "conserved" should be stated as "shared in two or more genes."
Pg. 10. The graphical depictions in Fig. 6 could be improved. There are many colored domains but no key to show what they are. Second the diagrams look like primary structures connected by even sized spacer but really aren’t. The original figure this was adapted from was better in this regard as it didn’t attempt to show connections between the domains.
P.11
At the last line of the third paragraph on page 11, "Hanschen et al., 2014" should be 2016.
P.13
On Figure 7, a red line and arrow depict putative lateral gene transfer. It should be made clear the diagram is a cladogram and not a chronogram. If it were the the latter, then the LGT was accompanied by time travel. Also, what reason is there to think that the direction of a possible LGT was from algae to metazoans and not the other way around?
P.16
On Page 16, the third line of the second paragraph, "2020a" but no other König & Nedelcu cited, so it should be just "2020."
PP.14-21
I have a problem with the conceptualization of a survival-reproduction trade-off dichotomy for gene function. Many or most genes are pleiotropic and are needed in some capacity for both survival and reproduction. Even “stress genes” like HSPs and heat shock factor have growth and reproductive functions under non-stress conditions. Case in point, the rls1 mutant of Chlamydomonas has pleiotropic phenotypes even under non-stress conditions (Saggere et al 2022), and is thus not strictly a life history tradeoff gene. Generally speaking, genes that participate in nutrient homeostasis are likely to have stress survival phenotypes while at the same time participate in growth and reproduction. A more nuanced conceptualization of this duality and a more cautious interpretation of existing data about regA and RLS1 function from published data—which is limited in scope and open to some serious caveats--would not take away from the ideas being presented.
P.18, ll.1-2
The phrase "the conditions that induced regA in regA mutant cells lacking a functional RegA protein" is phrased in an overly complicated manner. The conclusions from the Konig 2022 paper discussed in this section also should be interpreted more cautiously as the glsA regA double mutant strain has a different growth and development cycle from wild type, and the “control” experiment showing regA expression pattern in Eve did not look like it did in a previously published study of regA expression and in unpublished data that has been presented previously and even cited by the authors on pg. 7. If the authors believe that study captured the true expression pattern of regA, then they would need to explain why its highest level of expression corresponds to a time in development when the mutant has the most normal phenotype for its somatic cells, and why that gene is almost non-expressed later in development when the regA phenotype begins to show up.
P.18, ll.14-15
The current understanding that regA gene is not expressed in gonidia is mentioned even in König and Nedelcu 2020. I did not find this description in that article.
Pg. 20, 4th paragraph. “. . . due to the lack of an inversion process during development”. This is a confusing sentence that could be rephrased with some context to describe how embryonic polarity is attained and how cleavage and morphogenesis is different in Astrephomene than in other volvocine genera. As written, it implies that there is a fundamental difference in somatic cell placement with respect to embryonic polarity which may not be the case.
Reviewer 2 Report
This review is very interesting and will contribute much to our understanding molecular functional evolution of “germ-soma gene regA” in Volvox caarteri f. nagariensis. I have the following comments/questions.
1. The authors explain the distribution and evolution of VAL gene family homologs within the volvocine lineage by using very specific terminologies such as “Volvocaceae”, “Eudorina-group”, Goniaceae etc. However, broad readers would not understand the explanations in the text especially because figures lack such terminologies.
2. Similarly, the authors explain many multicellular volvocine genera in the text, but broad readers cannot understand “Platydorina” because no explanation in the text. Thus, summarized features of cell numbers, presence or absence of somatic cells, and colony shape should be shown in a table or diagrams in Figure 2.
3. The readers will consider why only Volvox carteri f. nagariensis is a model organism within the multicellular members of the volvocine algae, and why the function of VAL gene family homologs is known only in regA from Volvox carteri f. nagariensis, and why functions of VAL gene family homologs in other multicellular members are unknown. The authors should explain these questions in the text and propose future studies revealing the functions of the homologs of other multicellular organisms.
4. Figure 7 should clearly indicate “cryptophytes”, which is a small lineage.
5. Figure 4 should include the distinguishment of domains between C. reinhardtii and V. carteri.
6. Show the origin of Figure 3.
7. Page 6, 1st paragraph: This process is difficult to understand for general readers; Provide diagrams/photos for explanation.
8. Page 11: Cite the figure(s) [revised and/or new] to let the broad readers understand the VARL domain called “Pandorina’s Box” and a second short conserved motif, called the “PRL” motif after its conserved sequence, downstream of the VARL domain
